# Assessment and determinants of acute post-caesarean section pain in a tertiary facility in Ghana

Wisdom Klutse Azanu[1☯‡], Joseph Osarfo[2☯‡*], Roderick Emil Larsen-Reindorf[3,4], Evans Kofi Agbeno[5], Edward Dassah[4], Anthony Ofori Amanfo[5], Anthony Kwame Dah[1], Gifty Ampofo[2]

1 Department of Obstetrics and Gynaecology, School of Medicine, University of Health and Allied Sciences, Ho, Ghana, 2 Department of Community Medicine, School of Medicine, University of Health and Allied Sciences, Ho, Ghana, 3 Department of Obstetrics and Gynaecology, School of Medicine and Dentistry, Kwame Nkrumah University of Science and Technology, Kumasi, Ghana, 4 Department of Population, Family and Reproductive Health, School of Public Health, Kwame Nkrumah University of Science and Technology, Kumasi, Ghana, 5 Department of Obstetrics and Gynaecology, School of Medical Sciences, University of Cape Coast, Cape Coast, Ghana

☯ These authors contributed equally to this work.
‡ WKA and JO are joint first authors on this work.
* josarfo@uhas.edu.gh, josarfo@yahoo.co.uk

**Data Availability Statement:** All relevant data are within the paper and its Supporting Information files.

**Funding:** The author(s) received no specific funding for this work.

## Abstract

### Introduction

Caesarean sections (CS) feature prominently in obstetric care and have impacted positively on maternal / neonatal outcomes globally including Ghana. However, in spite of documented increasing CS rates in the country, there are no studies assessing the adequacy of post-CS pain control. This study assessed the adequacy of post-CS pain management as well as factors influencing this outcome. Additionally, post-CS analgesia prescription and serving habits of doctors and nurses were also described to help fill existing knowledge gaps.

### Methods

Pain scores of 400 randomly selected and consenting post-CS women at a tertiary facility in Ghana were assessed at 6–12 hours post-CS at rest and with movement and at 24–36 hours post-CS with movement using a validated visual analog scale (VAS) from February 1, 2015 to April 8, 2015. Participant characteristics including age, marital status and duration of CS were obtained using pretested questionnaires and patient records review. Descriptive statistics were presented as frequencies and proportions. Associations between background characteristics and the outcome variables of adequacy of pain control at 6–12 hours post-CS at rest and with movement and at 24–36 hours post-CS with movement were analysed using Chi-square and Fisher's exact tests and logistic regression methods. Adequate pain control was defined as VAS scores ≤5.

**Competing interests:** The authors have declared that no competing interests exist.

## Results

At 6–12 hours post-CS (at rest), equal proportions of participants had adequate and inadequate pain control (50.1% vrs 49.9%). Over the same time period but with movement, pain control was deemed inadequate in 93% of respondents (369/396). Women who had one previous surgery [OR 0.47 95%CI 0.27, 0.82; p = 0.008] and those whose CS lasted longer than 45 mins [OR 0.39 95% CI 0.24, 0.62; p<0.001] had lower odds of reporting adequate pain control. Women prescribed 12-hourly and 8-hourly doses of pethidine had only 23.5% (12/51) and 10.3% (3/29) served as prescribed respectively. At 24–36 hours post CS, adequate pain control was reported by 85.3% (326/382) of participants.

## Conclusions

Pain management was deemed inadequate within the first 12 hours post-CS with potential implications for early mother-child interaction. Appreciable numbers of participants did not have their analgesics served as prescribed. Adjunct pain control measures should be explored and healthcare workers must be encouraged to pay more attention to patients' pain relief needs.

## Introduction

Post-caesarean section (CS) pain, resulting from surgical tissue injury, is an important source of patient dissatisfaction and needs to be addressed aggressively for mothers to functionally recover quickly for optimization of the early stages of mother-child interaction [1–5]. Inadequately-controlled acute post-CS pain is associated with long hospital stay, increased costs and incidence of chronic pain [6–8]. Effective pain management is a benchmark for adequate health care and with CS being the most common surgical procedure conducted in the world [9, 10], healthcare providers must achieve adequate post-CS pain control as early as possible.

There is no 'gold standard' for post-CS pain management. Options are partly determined by drug availability, individual preferences, resource limitations and financial considerations and mostly rely on opioids, supplemented with anti-inflammatory analgesics, nerve blocks or other adjunctive techniques [5, 11, 12]. Factors such as the use of general anaesthesia, under-treatment with opioid analgesics fueled by fear of addiction or respiratory depression, the ability to request for more pain relief, pain threshold, religion, anxiety and pain anticipation are known to influence acute post-operative pain following CS and other surgeries [13–19]. Worsening of pain with movement compared to 'at rest 'position [20] demonstrates the need to achieve adequate pain control early in post-CS patients to facilitate early mother-child interaction including breastfeeding.

Caesarean sections are common in Ghana and national rates have increased from 9.8% in 2003 to 16%-18.5% in 2014 [21–23]. Eighty percent (80%) of about 90,000 obstetric and gynaecological surgeries performed in Ghana over 2014–2015 were caesarean sections [24]. In spite of these numbers, there is no known study on postoperative pain assessment conducted solely on post-CS patients in Ghana. One study [25] sought to validate pain scales in postoperative patients, majority of whom were post-CS but did not report on the burden of acute post-CS pain. Furthermore, availability of options for post-CS analgesia implies use of varying analgesics and dosing regimen. This situation could allow considerable variation in pain management with potential implications for quality of care.

The study primarily sought to assess the adequacy of pain management post-CS, using a validated pain scale, from the perspective of women who had CS done in the second largest tertiary facility in Ghana as well as factors influencing this outcome. Secondarily, the prescription and serving habits of doctors and nurses respectively for post-CS analgesia were assessed.

## Materials and methods

### Study design, study area and population

The study employed a cross-sectional design with repeated measures at fixed time intervals to assess changes in the perceived adequacy of post-CS pain control. It was conducted at the lying-in wards of the Obstetrics and Gynaecology Directorate of the Komfo Anokye Teaching Hospital (KATH) from February 1, 2015 to April 8, 2015."

The study population comprised women who had had CS done at KATH with pain assessment up to 36 hours post-CS. The hospital (KATH) is located in the Ashanti Region in the middle belt of Ghana and serves as a referral center for the entire northern half and some parts of southern Ghana. Close to 12,000 deliveries are conducted annually and about a third of these are by CS (Biostatistics Unit, KATH, 2014).

### Sample size determination

The sample size was estimated using the Cochran's formula; n = $(z/m)^2 P(1-P)$ where $z$ is the reliability coefficient ie 1.96 at 95% confidence interval, $p$ is the estimated proportion of post-CS women with inadequate pain and $m$ is precision. Assuming a 37% prevalence of post-CS women with inadequate pain control from a previous study [1] and a 5% error margin, a sample size of 359 was estimated. Adding 10% for non-response and data that may not contribute to analysis, a final sample size of 400 was used.

### Participant selection, study procedures and data collection tools

Post-CS women of all ages in the lying-in wards at KATH were invited to participate in the study. Women were deemed eligible if they (i) had their CS done at KATH and (ii) were conscious and communicating by 6–12 hours after the surgery. No consideration was given to their physical status pre-surgery. For clients below 18 years, assent was obtained from the client and consent obtained from the attending guardian. Participants were excluded if there was a language barrier which could challenge communication. Information was extracted from the theatre register for the purposes of participant selection while socio-demographic, clinical characteristics and pain scores were obtained using patient folders and questionnaires after obtaining informed consent.

The theatre register was reviewed at 12-hourly intervals to identify potential study participants. At each theatre register review, 3 clients were selected by simple random sampling using all the women who had had CS in the preceding 12 hours as the sample frame. The selected clients were then assessed for eligibility. Where a selected client did not meet eligibility criteria, another client who had also had CS in the preceding 12 hours was selected randomly from the theatre register as replacement. Patient folders, including anaesthetic records and drug administration charts, were also reviewed to extract data on whether the CS was planned or an emergency, the grade of surgeon, duration of the CS, types of skin incision and analgesia used, the prescribed analgesics and whether they were served by the nurses as prescribed. No defined protocol or choice of medicines for post-CS analgesia existed at the time of study conduct and analgesia prescription was driven by availability and individual surgeon preference. The

questionnaire captured data on participant socio-demographic and surgery-related characteristics such as age, level of education, marital status and history of previous surgery among others.

The questionnaire also included a visual analog scale (VAS) for scoring the post-CS pain intensity (see S1 Appendix). The VAS has been validated and allows a reliable and consistent measure of pain intensity [26] with scores from 0 to 10; 10 representing the most severe pain. The pain scores by the VAS were categorized to define pain control as 'adequate pain control' (0–5) and 'inadequate pain control' (6–10). The middle score on the VAS was arbitrarily chosen as the border between adequate and inadequate pain control.

Pain assessment was done for each participant on two separate occasions; between 6–12 hours post-CS (representing pain score on the day of operation) and at 24–36 hours post-CS (representing pain score on post-operative day 1). Pain at 6–12 hours was assessed at rest (defined as lying in bed) and with movement (defined as elevation from the horizontal to the sitting position). Pain assessment was only done with movement at 24–36 hours post-CS as assessment at rest at this time was not deemed relevant. Pain assessment with body movement was deemed important because the expectation is for post-CS women to breastfeed as soon as possible and they need adequate pain control to effectively do so. The selected time intervals were chosen for the convenience of the investigators.

The questionnaire was administered in the local Asante Twi language by the lead author and two research assistants who had prior experience in data collection and had been trained to translate the questionnaire in a standardized manner. The questionnaire was pretested at the Kumasi South Hospital where 40 questionnaires were administered over seven days and subsequent appropriate revisions made.

## Data management and analysis

Data was double entered in SPSS version 16 (SPSS, IBM, USA) and exported into STATA 11 (Stata Corp, Texas, USA) for cleaning and analysis. Descriptive analysis of participant demographic and clinical characteristics (explanatory variables) and pain scores were conducted and presented as frequencies, proportions, percentages and means. Chi-square and Fisher's exact tests were conducted for association between the explanatory variables and the co-primary outcome variables of pain control adequacy at 6–12 hours post-CS (at rest and with movement) and the secondary outcome variable of adequate pain control at 24–36 hours. Logistic regression analysis was done to ascertain the strength of association for the independent variables showing significant association with the binary outcome variables. Associations between explanatory and dependent variables were deemed significant if p-value ≤0.05. The minimal dataset underlying the results described has been included as S3 Appendix.

## Ethical approval

The study was approved by the Committee on Human Research and Publication Ethics (CHRPE) of the School of Medical Sciences, Kwame Nkrumah University of Science and Technology and the Komfo Anokye Teaching Hospital with reference number CHRPE /AP/ 106/14. The management of KATH granted written permission to conduct the study in the hospital. Written informed consent was obtained from all eligible participants and confidentiality ensured by anonymizing them. For participants below 18 years of age (one aged 14, two aged 16 and three aged 17), assent was documented but a verbal consent was obtained from the parent or guardian. Verbal parental/guardian informed consent for such cases was specifically approved by the ethics committee before commencement of the study.

## Results

### Participant socio-demographic and clinical characteristics

Table 1 shows the background demographic and surgery-related characteristics of the study women. Majority of the women were in the age group 25–35 years (62.7%, 249/367) and were married (80.3%, 313/390). The mean age (SD) was 27.8 years (5.8). Close to half (48.9%, 194/393) were of parity ≥3. About two-thirds (66.8%, 261/391) of the CS were done as emergencies and 6 in 10 (60.5%, 210/347) were performed by residents. About half of the CS were completed in 45 minutes or less (49.6%, 193/389).

### Prescription and serving of analgesics

Pethidine, morphine and diclofenac were the analgesics prescribed for pain control in the first 24 hours post-CS (see Table 2). There were discrepancies in doses between prescriptions in the patient folders and what was administered by nurses as detected from drug administration chart reviews. While 226 women were prescribed intravenous pethidine 100mg statim, only 90.3% (204/226) were served the medication. Overall, women prescribed 12hourly and 8hourly doses of pethidine respectively had only 23.5% (12/51) and 10.3% (3/29) served as prescribed. Similarly, of the 361 women prescribed a 100mg 12hourly dose of suppository diclofenac, only 304 (84.2%) were served.

### Assessment of post-CS pain

At 6–12 hours post-CS (at rest), there were equal proportions of study women with adequate and inadequate pain control [50.1%, (199/397) vrs 49.9%, (198/397)]. Table 3 shows the association between the background demographic/ clinical characteristics and the adequacy of pain control at 6–12 hours post CS. Marital status (p = 0.028), parity (p = 0.039), previous surgery (p = 0.034), grade of surgeon (p = 0.042) and duration of the CS procedure (p<0.001) were significantly associated with adequacy of pain control at 6–12 hours post-CS at rest.

These variables were then entered in a logistic regression model to assess their strengths of association (see Table 4). Bivariate analysis showed women who were married [OR 2.26 95% CI 1.00, 5.13; p = 0.05] or unmarried but co-habiting with their sexual partners [OR 3.70 95% CI 1.39, 9.87; p = 0.009] were more likely to report adequate pain control 6–12 hours post-CS at rest compared to women who were single. Similarly, women who had their CS done by residents [OR 2.75 95% CI 1.08, 7.02; p = 0.035] and senior residents or higher grades of doctors [OR 3.33 95% CI 1.26, 8.81; p = 0.015] were also more likely to experience adequate pain control at 6–12 hours post-CS at rest compared to those whose CS were done by house officers. Pregnant women whose CS took longer than 45 minutes were less likely to report adequate pain control [OR 0.33 95% CI 0.22, 0.50; p<0.001].

In the multivariate analysis, only previous surgery and duration of CS retained significant association with adequate pain control 6–12 hours post-CS at rest. Women who had had one previous surgery [AOR 0.47 95%CI 0.27, 0.82; p = 0.008] and those whose CS lasted longer than 45 mins [AOR 0.39 95% CI 0.24, 0.62; p<0.001] had lower odds of reporting adequate pain control.

With movement at 6–12 hours post-CS, pain control was perceived to be inadequate in over 90% of respondents (369/396, 93.2%) and none of the background demographic and surgical characteristics was associated with adequacy of pain control (see Table 3). On the contrary, at 24–36 hours post-CS, adequate pain control was reported in 85.3% (326/382) of participants and only marital status (p = 0.010) was found to be significantly associated with pain control (see Table 5).

**Table 1. Demographic and surgical characteristics of the study participants.**

| Variable | Frequency | % |
|---|---|---|
| **Age (years) (N = 397)** | | |
| 14–24 | 72 | 18.2 |
| 25–35 | 249 | 62.7 |
| 36–46 | 76 | 19.1 |
| **Level of Education (N = 385)** | | |
| None | 45 | 11.7 |
| Primary/JSS | 212 | 55.1 |
| At least SHS | 128 | 33.3 |
| **Marital Status (N = 390)** | | |
| Single | 29 | 7.4 |
| Married | 313 | 80.3 |
| Co-habiting | 48 | 12.3 |
| **[a]Occupation (N = 394)** | | |
| Unemployed | 49 | 12.4 |
| Unskilled worker | 193 | 49.0 |
| Skilled worker | 152 | 38.6 |
| **Parity (N = 393)** | | |
| 1 | 94 | 23.9 |
| 2 | 107 | 27.2 |
| ≥3 | 192 | 48.9 |
| **Previous surgery (N = 392)** | | |
| None | 216 | 55.1 |
| 1 | 92 | 23.5 |
| ≥2 | 84 | 21.4 |
| **Nature of CS (N = 391)** | | |
| Emergency | 261 | 66.8 |
| Plannned / Elective | 130 | 33.2 |
| **[b]Grade of Surgeon (N = 347)** | | |
| Houseofficer | 22 | 6.4 |
| Resident | 210 | 60.5 |
| Senior Resident and higher | 115 | 33.1 |
| **Duration of CS (mins) (N = 389)** | | |
| ≤ 45 mins | 193 | 49.6 |
| >45 mins | 196 | 50.4 |
| **Skin Incision (391)** | | |
| Midline | 22 | 5.6 |
| Pfannenstiel | 369 | 94.4 |
| **Anaesthesia (N = 392)** | | |
| Spinal Block | 379 | 96.7 |
| General | 13 | 3.3 |

[a]unskilled worker included the likes of subsistence farmers, traders, domestic workers, etc while skilled workers included hairdressers, seamstresses, public servants, etc.

[b]Houseofficer comprised first and second-year houseofficers while the higher ranks included senior specialists and consultants.

**Table 2. Prescription and administration of analgesia for 24 hours post-CS.**

| Analgesic and dosage regimen | Number of women analgesic was prescribed for n (%) | Number of women served n (%) |
|---|---|---|
| **Pethidine** | | |
| 100mg st | 226 (73.4) | 204 (93.1) |
| 100mg 12-hourly | 51 (16.6) | 12 (5.5) |
| 100mg 8-hourly | 29 (9.4) | 3 (1.4) |
| 100mg 6-hourly | 2 (0.6) | 0 (0.0) |
| Total | 308 (100) | 219 (100) |
| **Diclofenac** | | |
| 100mg 12-hourly | 361 (96.3) | 304 (96.8) |
| 75mg 12-hourly | 14 (3.7) | 10 (3.2) |
| Total | 375 (100) | 314 (100) |
| **Morphine** | | |
| 10mg st | 25 (65.8) | 19 (79.2) |
| 10mg 12-hourly | 8 (21.0) | 3 (12.5) |
| 10mg 8-hourly | 5 (13.2) | 2 (8.3) |
| Total | 38 (100) | 24 (100) |

Logistic regression analysis showed that married women (OR 3.12 95%CI 1.31, 7.40; p = 0.010) and those cohabiting (OR 7.00 95%CI 1.70, 28.9; p = 0.007) still had higher odds of reporting adequate pain control at 24–36 hours post-CS compared to single women (not shown in any table). Adjusting for parity, history of previous surgery, grade of surgeon and duration of CS, married women (AOR 3.49 95%CI 0.94, 12.98; p = 0.063) and cohabiting women (AOR 5.03 95%CI 0.95, 26.71; p = 0.058) had higher odds of adequate pain control at 24–36 hours though this was not statistically significant. Backward elimination of the potential confounders leaving only parity and history of previous surgery showed married women (AOR 3.26 95%CI 1.32, 8.07; p = 0.011) and those cohabiting (AOR 7.58 95%CI 1.8, 31.56; p = 0.005) remained more likely to report adequate pain control (see S2 Appendix).

## Discussion

We assessed the adequacy of post-CS analgesia to help fill gaps in this area of knowledge in Ghana. This report is the first to describe the burden of inadequate post-caesarean section pain with emphasis on disparities between scores at rest and with movement in Ghana. Almost half of the participants had inadequate pain control 6–12 hours after the procedure even when lying down. Notably, significant numbers of participants did not have their analgesics served to them as prescribed by the time of review of patient records.

Inadequate pain control was reported by 9 out of 10 women in the study at 6–12 hours post CS with movement. Movement, for any purpose including breastfeed, increased pain and erased associations found at rest within the same time period. Moving from a lying position to sitting in bed will likely involve some stretching of abdominal muscle and tissue that generate stimuli interpreted as pain. This observation is similar to reports of high frequencies and levels of pain with movement among post-CS women in Brazil [4, 27]. It is worrisome that over 90% of the mothers still perceived inadequate pain control with movement at this time as it can interfere with early bonding. Furthermore, it raises questions about the adequacy of the post-CS analgesia used. This observation could be consequent to nurses' not appropriately serving pain medication as prescribed as was observed in the present study. Nurses on the ward must better appreciate the challenges posed by prolonged inadequate pain control in post-CS

**Table 3. Assessment of association between participants' demographic and clinical characteristics and adequacy of pain control at 6–12 hours post-CS.**

| @Variable | 6–12 hours (rest) | | | 6–12 hours (movement) | | |
|---|---|---|---|---|---|---|
| | Adequate Pain control n (%) | Inadequate Pain control n (%) | p-value | Adequate Pain control n (%) | Inadequate Pain control n (%) | *p-value |
| **Age (years) (N = 397)** | | | 0.559 | | | 0.489 |
| 14–24 | 36 (18.1) | 36 (18.2) | | 3 (11.1) | 69 (18.7) | |
| 25–35 | 129 (64.8) | 120 (60.6) | | 20 (74.1) | 228 (61.8) | |
| 36–46 | 34 (17.1) | 42 (21.2) | | 4 (14.8) | 72 (19.5) | |
| **Education (N = 385)** | | | 0.953 | | | 0.742 |
| None | 22 (11.2) | 23 (12.2) | | 2 (7.4) | 42 (11.8) | |
| Primary/JSS | 109 (55.6) | 103 (54.5) | | 17 (63.0) | 195 (54.6) | |
| At least SHS | 65 (33.2) | 63 (33.3) | | 8 (28.6) | 120 (33.6) | |
| **Marital Status (N = 390)** | | | **0.028** | | | 0.073 |
| Single | 9 (4.6) | 20 (10.4) | | 2 (7.4) | 27 (7.5) | |
| Married | 158 (80.2) | 155 (80.3) | | 25 (92.6) | 287 (79.3) | |
| Co-habiting | 30 (15.2) | 18 (9.3) | | 0 | 48 (13.2) | |
| **Occupation (N = 394)** | | | 0.641 | | | 0.853 |
| Unemployed | 22 (11.0) | 27 (13.9) | | 3 (11.1) | 46 (12.6) | |
| Unskilled worker | 101 (50.8) | 92 (47.1) | | 12 (44.4) | 181 (49.4) | |
| Skilled worker | 76 (38.2) | 76 (39.0) | | 12 (44.4) | 139 (38.0) | |
| **Parity (N = 393)** | | | **0.039** | | | 0.490 |
| 1 | 52 (26.7) | 42 (21.2) | | 4 (16.0) | 90 (24.5) | |
| 2 | 42 (21.5) | 65 (32.8) | | 6 (24.0) | 101 (27.5) | |
| ≥3 | 101 (51.8) | 91 (46.0) | | 15 (60.0) | 176 (48.0) | |
| **Previous surgery (N = 392)** | | | **0.034** | | | 0.894 |
| None | 122 (61.3) | 94 (48.7) | | 14 (51.9) | 202 (55.5) | |
| 1 | 38 (19.1) | 54 (28.0) | | 7 (25.9) | 85 (23.4) | |
| ≥2 | 39 (19.6) | 45 (23.3) | | 6 (22.2) | 77 (21.1) | |
| **Nature of CS (N = 391)** | | | 0.058 | | | 0.093 |
| Emergency | 141 (71.2) | 120 (62.2) | | 14 (51.9) | 247 (68.0) | |
| Plannned / Elective | 57 (28.8) | 73 (37.8) | | 13 (48.1) | 116 (32.0) | |
| **Grade of Surgeon (N = 347)** | | | **0.042** | | | 0.800 |
| Houseofficer | 7 (3.6) | 15 (9.9) | | 2 (7.4) | 20 (6.3) | |
| Resident | 118 (60.5) | 92 (60.5) | | 17 (63.0) | 193 (60.5) | |
| Senior Resident and higher | 70 (35.9) | 45 (29.6) | | 8 (29.6) | 106 (33.2) | |
| **Duration of CS (mins) (N = 389)** | | | **<0.001** | | | 0.690 |
| ≤ 45 mins | 124 (62.9) | 69 (35.9) | | 14 (53.9) | 179 (49.5) | |
| >45 mins | 73 (37.1) | 123 (64.1) | | 12 (46.1) | 183 (50.5) | |
| **Skin Incision (N = 391)** | | | 0.160 | | | 0.188 |
| Midline | 8 (4.0) | 14 (7.3) | | 3 (11.1) | 19 (5.2) | |
| Pfannenstiel | 191 (96.0) | 178 (92.7) | | 24 (88.9) | 344 (94.8) | |
| **Type of Anaesthesia (N = 392)** | | | 0.429 | | | 1.000 |
| Spinal Block | 191 (96.0) | 188 (97.4) | | 27 (100.0) | 351 (96.4) | |
| General | 8 (4.0) | 5 (2.6) | | 0 (0) | 13 (3.6) | |

@for all variables, the number of participants (N) is one less in those assessed for pain at 6–12 hours (movement)

*Fisher's exact p-values are reported for 6–12 hours (movement).

**Table 4. Logistic regression analysis output for association with pain scores at 6–12 hours post-CS (rest).**

| Variable | Crude OR (95% CI) | p-value | *Adjusted OR (95% CI) | p-value |
|---|---|---|---|---|
| **Marital Status** | | | | |
| $^\$$Single | 1 | | 1 | |
| Married | 2.26 (1.00, 5.13) | **0.05** | 1.37 (0.49, 3.85) | 0.546 |
| Co-habiting | 3.70 (1.39, 9.87) | **0.009** | 1.86 (0.58, 5.94) | 0.292 |
| **Parity** | | | | |
| 1 | 1 | | 1 | |
| 2 | 0.52 (0.30, 0.92) | **0.023** | 0.69 (0.36, 1.31) | 0.256 |
| ≥3 | 0.90 (0.55, 1.47) | 0.665 | 1.36 (0.74, 2.53) | 0.325 |
| **Previous Surgery** | | | | |
| None | 1 | | 1 | |
| 1 | 1.84 (1.12, 3.02) | **0.015** | 0.47 (0.27, 0.82) | **0.008** |
| ≥2 | 1.23 (0.68, 2.24) | 0.494 | 0.61 (0.32, 1.17) | 0.139 |
| **Grade of Surgeon** | | | | |
| Houseofficer | 1 | | 1 | |
| Resident | 2.75 (1.08, 7.02) | **0.035** | 2.51 (0.94, 6.67) | 0.066 |
| Senior Resident or higher | 3.33 (1.26, 8.81) | **0.015** | 2.71 (0.96, 7.67) | 0.060 |
| **Duration of CS** | | | | |
| ≤ 45 mins | 1 | | 1 | |
| > 45 mins | 0.33 (0.22, 0.50) | **<0.001** | 0.39 (0.24, 0.62) | **<0.001** |

*Each variable was adjusted for the other variables shown

$^\$$the first category of each variable was the reference group.

women. Regular professional development programmes in pain management are needed for them to become more receptive to patient needs. The observation also makes a case for exploration of patient-controlled analgesia in our local set-up. It would have been insightful to relate adequacy of pain control to the numbers of women who may have received a single analgesic compared to combined analgesics but this was not explored in the current study. While it is highly unlikely that any participant was prescribed a single analgesic, we cannot rule out the possibility that some may have been served a single analgesic.

The time ranges for assessment of pain scores in the present study limits comparison to other studies that assessed pain at specific points in time. Nevertheless, the findings show post-CS pain management at KATH was reported inadequate by 12 hours post-CS and compared well to a Norwegian study [28] in which 68% of participants had inadequate pain control 24 hours after CS. Presumably, the Norwegian study would have had a higher proportion with inadequate pain control at 12 hours post-CS. Despite the high prevalence of inadequate pain control at 6–12 hours with movement in the present study, it was observed during data collection that all the women still breastfed their infants. Though this was not documented as an outcome in the results, it is worthy of note and agrees with the assertion that pain-rating scores may not exactly measure physical comfort and independence [10]. It is also possible that maternal instincts or an awareness of the advantages of continuously putting the child to breast drove the women to still breastfeed in spite of their high pain scores within this time period.

The analgesics prescribed were similar to those used in a Ugandan tertiary facility [29]. The present study observed that for every 10 women prescribed a 100mg statim dose of pethidine, one did not get served by the attending nurses. The situation was worse among those prescribed multiple doses. Patients not getting the full complement of their analgesia is a common occurrence after surgical procedures including CS [29–33]. The current study did not explore

**Table 5. Association between participant demographic and clinical characteristics and adequacy of pain control at 24–36 hours post-CS.**

| Variable | Adequate Pain Control n (%) | Inadequate Pain Control n (%) | p-value |
|---|---|---|---|
| **Age (N = 382)** | | | 0.444 |
| 14–24 | 61 (18.7) | 9 (16.1) | |
| 25–35 | 199 (61.0) | 39 (69.6) | |
| 36–46 | 66 (20.3) | 8 (14.3) | |
| **Education (N = 371)** | | | *0.351 |
| None | 39 (12.2) | 3 (5.8) | |
| Primary/JHS | 170 (53.3) | 32 (61.5) | |
| At least SHS | 110 (34.5) | 17 (32.7) | |
| **Marital Status(N = 376)** | | | *__0.010__ |
| Single | 18 (5.6) | 9 (16.7) | |
| Married | 262 (81.4) | 42 (77.8) | |
| Co-habiting | 42 (13.0) | 3 (5.5) | |
| **Occupation (N = 379)** | | | 0.340 |
| Unemployed | 39 (12.0) | 7 (12.7) | |
| Unskilled | 163 (50.3) | 22 (40.0) | |
| Skilled | 122 (37.7) | 26 (47.3) | |
| **Parity (N = 378)** | | | 0.862 |
| Para 1 | 76 (23.6) | 15 (26.8) | |
| Para 2 | 88 (27.3) | 14 (25.0) | |
| Multipara | 158 (49.1) | 27 (48.2) | |
| **Previous surgery (N = 378)** | | | 0.921 |
| None | 179 (55.2) | 31 (57.4) | |
| 1 | 76 (23.5) | 13 (24.1) | |
| $\geq$2 | 69 (21.3) | 10 (18.5) | |
| **Nature of CS (N = 377)** | | | 0.869 |
| Emergency | 219 (67.8) | 36 (66.7) | |
| Planned/Elective | 104 (32.2) | 13 (33.3) | |
| **Grade of Surgeon(N = 336)** | | | *0.463 |
| Houseofficer | 19 (6.3) | 2 (6.1) | |
| Resident | 187 (61.7) | 17 (51.5) | |
| Senior Resident and higher | 97 (32.0) | 14 (42.4) | |
| **Duration of CS(N = 376)** | | | 0.052 |
| $\leq$ 45 mins | 168 (52.3) | 21 (38.2) | |
| > 45 mins | 153 (47.7) | 34 (61.8) | |
| **Skin Incision(N = 390)** | | | *0.092 |
| Midline | 20 (6.2) | 0 | |
| Pfannenstiel | 303 (93.8) | 54 (100.0) | |
| **Anaesthesia (N = 391)** | | | *0.228 |
| Spinal | 311 (96.3) | 55 (100.0) | |
| General | 12 (3.7) | 0 | |

*Fisher's exact p-values reported.

nurses' non-adherence to prescriptions but fear of addiction may underlie this observation for pethidine in particular as has been previously described in Ghana [17]. It is also possible that the nurses' perception of patients' pain was at variance with the patients' own pain perception and thus they may not have seen the need to administer the subsequent doses of the pethidine. Such discrepant perceptions of pain were reported in an earlier study of non-obstetric post-

operative pain management that was conducted 5 years prior to the present study in the same teaching hospital in Ghana [34].

In cases where multiple doses of opioids are prescribed, the observed 'practice' in the facility is for the pharmacy unit to serve the medication when the patient is due. This is done to avert potential opioid abuse and addiction among healthcare workers and is founded on previous occurrence of such behaviour. The attending nurses thus have to go for the prescribed doses in 'bits' and it appears there may be some challenges with this practice. Ensuring individualised nursing care with a clear-cut care plan at the outset of each nurse's duty will ensure that the needed supplies for the best care of that patient are sought for well ahead of time to overcome this hurdle.

Women who had one previous surgery and women whose CS lasted longer than 45 minutes had lower odds of reporting adequate pain control at 6–12 hours post-CS at rest. Inadequate pain control in those with one previous surgery compared to those with no previous surgery aligns with reports that previous surgical history may increase patients' pain sensitivity [35, 36]. This could stem from adhesions formation and the potential increased nociceptor sensitivity resulting from more tissue handling and/or damage where the CS takes longer.

Invariably, women with more than one previous surgery would be expected to have more acute postoperative pain but contrary observations were made in the present study. It is not clear why women with ≥2 previous surgeries had less postoperative pain than those with only one previous surgery. One would expect women with ≥2 previous surgeries to have more adhesions and hence more pain compared to those with fewer surgeries. It is highly unlikely that all the women with ≥2 previous surgeries either did not have adhesions at all or had rather minimal levels of it. In a further effort to understand the phenomenon, we cross-tabulated the variables "previous surgery" and "grade of surgeon". Houseofficers, who are generally inexperienced and thus more likely to handle tissues with less finesse irrespective of working under supervision, performed the CS on less than 10% of women with no previous surgery (15/182) and one previous surgery (7/89) respectively. All those with ≥2 previous surgeries were handled by residents, senior residents and other higher grades of doctors. It would thus appear the observation has little to do with the experience of the surgeon. It is also possible women with ≥2 previous surgeries may have somehow mentally prepared themselves to cope with the pain better than those with one previous surgery.

Married women and those unmarried but co-habiting with their sexual partners had higher odds of reporting adequate pain control at 6–12 hours post-CS at rest compared to single women. The influence of marriage and cohabitation was also noted at 24–36 hours post-CS but this must be interpreted with caution as the grade of the surgeon performing the procedure may be confounding, albeit weakly and the confidence intervals of the adjusted odds ratios were rather wide despite having significant p-values.

While the stability of their relationships was not assessed, it is possible married women and those cohabiting have better bonding and social support from their partners with subsequent less personal anxiety. Single women are more likely to have less stable relationships where their partners can deny responsibility for the pregnancy. Also, getting pregnant outside a clearly-defined marriage environment may attract some societal stigma in our local setting and predispose to stress and anxiety. Pre-operative anxiety has been linked to moderate / severe acute post-CS pain [27, 37]. This apparent spousal / partner support has been identified to influence other aspects of pregnancy as studies, including a Ghanaian one, have reported cohabiting women are more likely to attend antenatal clinics (ANC) while the lack of partner support is associated with late ANC attendance [38, 39].

The strength of the study lies in the assessment of adequate pain management at rest and with movement. Secondly, the random selection of participants assures that the study sample

was representative of women who had undergone CS at KATH. The study has limited generalizability as it is fully hinged on post-CS analgesia given at KATH. However, because these may not be markedly different from that used in other hospitals in the country, the study findings can be said to appreciably highlight the challenges of inadequate pain control 12 hours post-CS and beyond in Ghana and the implications for mothers as they need to breastfeed often at this time to help fully establish lactation.

In addition, pharmacokinetic assessments would have enabled comparison of blood analgesic levels with pain perception but this was not done. The study findings, however, remain valid as a validated pain score scale was used and set the background for a second look at post-CS analgesia in Ghana.

## Conclusion

There were apparent challenges with the adequacy of acute post-CS pain management at KATH as over 90% of participants reported inadequate pain control 6–12 hours post-CS with movement from the 50% reported at rest for the same time duration. The study also showed evidence of discrepancy between doses of analgesics prescribed and what was served to participants. The limited adequacy of pain control within the first 12 hours post-CS with movement is of concern and could have adverse implications for early mother-child engagement. This can be further compounded by the observation of nurses' not appropriately serving analgesics as prescribed. Additional options for post-CS pain control such as patient-controlled analgesia or abdominal plane nerve blocks may have to be explored in our set-up. Nurses on the wards have to be updated on post-operative pain management. Doctors may have to frequently check on patient drug administration to ensure maximum compliance with serving but this may also be a burden on their hectic workloads. Further studies are needed to elucidate health system factors that may be contributing to the problem including current reasons for nurses 'failing' to serve medications as prescribed. The possible interference of post-CS pain with breastfeeding and mother-baby bonding, however short-lived, may also be explored in future studies by collecting data on periods of breastfeeding and using a bonding scale. Also, the influence of other factors such as gestation, length of surgical incision and level of sensory block of the spinal anaesthesia can be further investigated in new studies that utilize parameters like total post-operative analgesic consumption in addition to the VAS for better assessment of the effectiveness of pain control.

## Supporting information

**S1 Appendix. Visual analog scale (VAS).**
(DOCX)

**S2 Appendix. Stata bivariate and multivariate logistic regression analysis output.**
(DOCX)

**S3 Appendix. Assessment of post CS data.**
(XLS)

## Acknowledgments

We are grateful to the study women, the doctors and nurses at the obstetrics and gynaecology directorate of KATH, the management of KATH and the research team.

## Author Contributions

**Conceptualization:** Wisdom Klutse Azanu, Roderick Emil Larsen-Reindorf, Evans Kofi Agbeno, Edward Dassah.

**Data curation:** Wisdom Klutse Azanu, Joseph Osarfo, Edward Dassah.

**Formal analysis:** Joseph Osarfo.

**Investigation:** Wisdom Klutse Azanu.

**Methodology:** Wisdom Klutse Azanu, Roderick Emil Larsen-Reindorf, Evans Kofi Agbeno, Edward Dassah.

**Project administration:** Wisdom Klutse Azanu.

**Visualization:** Joseph Osarfo, Gifty Ampofo.

**Writing – original draft:** Wisdom Klutse Azanu, Joseph Osarfo, Evans Kofi Agbeno, Gifty Ampofo.

**Writing – review & editing:** Wisdom Klutse Azanu, Joseph Osarfo, Roderick Emil Larsen-Reindorf, Evans Kofi Agbeno, Edward Dassah, Anthony Ofori Amanfo, Anthony Kwame Dah, Gifty Ampofo.

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
