## [Decision Letter · Decision Letter 0]

9 Dec 2021

PONE-D-21-20342Assessment of the burden and factors influencing acute post-caesarean section pain in a tertiary facility in Ghana.PLOS ONE

Dear Dr. Osarfo,

Thank you for submitting your manuscript to PLOS ONE. After careful consideration, we feel that it has merit but does not fully meet PLOS ONE’s publication criteria as it currently stands. Therefore, we invite you to submit a revised version of the manuscript that addresses the points raised during the review process.

The reviewers have raised a number of concerns that we feel should be addressed. They feel that the portions of the methodology should be more clearly described. They also feel that the reliability of the results is questionable and this should be addressed. The reviewers also noted a number of English grammar and language issues and typos that must be resolved. The reviewers' comments can be viewed in full, below.

We look forward to receiving your revised manuscript.

Kind regards,

Natasha McDonald, PhD

Associate Editor

PLOS ONE

Journal Requirements:

2. In the ethics statement please indicate the type of informed consent obtained from the parent or guardian of the minors included in the study (ie written, verbal).

Reviewers' comments:

Reviewer's Responses to Questions

**Comments to the Author**

1. Is the manuscript technically sound, and do the data support the conclusions?

Reviewer #1: Yes

Reviewer #2: No

2. Has the statistical analysis been performed appropriately and rigorously? 

Reviewer #1: Yes

Reviewer #2: No

3. Have the authors made all data underlying the findings in their manuscript fully available?

Reviewer #1: Yes

Reviewer #2: Yes

4. Is the manuscript presented in an intelligible fashion and written in standard English?

Reviewer #1: Yes

Reviewer #2: Yes

5. Review Comments to the Author

Reviewer #1: PlosONE Review – CS pain control in KATH, Ghana

Typos:

‘pan’ instead of ‘pain, p3

Explanations:

Clear and transparent explanations for everything that was carried out including precise definitions of adequate or inadequate pain control, ‘at rest’ and ‘with movement’, and a validated pain score scale used within the questionnaire.

Statistics:

Data analysis described clearly and conducted with precision.

Results - interpretations:

As there were significant differences in experiences of pain for married or cohabiting women, this indicates that women need emotional support for pain management and/or someone to call out for their pain control needs.

Content relevance:

Well-constructed, original research that fills a gap in the literature around pain control after CS in Ghana.

Future research:

Could establish what can be done about the discrepancy between prescribed and administered analgesia by the nurses – e.g., as the authors suggested, further training in the importance of pain management for patient wellbeing and potentially the early mother-infant relationship. As the authors point out, future research could also measure differences in experienced pain between women who received a single analgesic and those who received combined analgesics. In addition, breastfeeding outcomes could be monitored and recorded to test the presumed associations between level of pain and ability to initiate successful breastfeeding.

Issues:

1. P19 – movement was considered ‘presumably to breastfeed’ although women could move for other reasons post CS, such as visiting the bathroom, reaching for something such as a tissue or glass of water, or simply sitting up to eat a meal.

2. The authors speak about the possible interference of participant pain with mother-baby bonding, yet they do not mention using a bonding scale. Therefore, this gap could be filled in follow up work or in some future research study.

3. Although they state that women breastfed their babies despite inadequate pain control, they do not specify exactly how many women managed to do this or if it was all the participants. Although, as traditionally 100% of women initiate breastfeeding in some African countries such as Tanzania, this may have been the case in Ghana too. This could be clarified for western readers in the discussion on p20.

Summary

The main focus on the conclusion was on the findings about the difference in prescribed and administered pain control medication. Another important point seemed to be the difference in pain for women who received more social support from a partner and those who did not. It could also be that their pain control needs were better supported through having their partner present to speak up for them. It might be a good idea to state that a follow up study could benefit from the inclusion of validated mother-infant interactions and/or bonding scales, and also breastfeeding data, especially given that the authors’ main concern appeared to be the impacts of experienced pain and lack of effective pain control on mother-infant bonding and breastfeeding.

Reviewer #2: In general the area of focus of the article is very interesting. But the papers requires major revision in the following areas

Title

The title and the objectives of the study are not inline. For instance, the title included the burden of pain management but the main body of the document showed nothing regarding the burden of poor post cesarean delivery pain management.

Abstract

The introduction and method sessions of the abstract didn’t provide clear information about the problem and the methodology the researchers went through

Introduction

Well written, but in the objective part terms like adequacy are not measurable and vague

Methodology

In general the methodology session should address at least the following points

• The study design is not clearly described. It simply says prospective study. There are different prospective study types

• It would be good if the post operative pain management protocol of the hospital was described.

• The inclusion and exclusion criterias are not clearly stated. For instance, giving consent to participate in the study should not be considered as inclusion criteria. What about the American Society of Anesthesiologists (ASA) physical status of the patients for inclusion and exclusion criteria? Other possible inclusion and exclusion should be addressed.

• The study didn’t consider other important factors which influence the severity of post cesarean delivery pain. For example, gestational age, type of local anesthetic used for spinal anesthesia, type of the post operative pain management modality used ( regional analgesia vs systemic analgesics), length of surgical incision, level of sensory block of the spinal anesthesia, quality of intraoperative analgesia and etc. were not addressed in the study. These factors are very critical for post cesarean delivery pain and analgesic quality assessment

• Other important parameters to assess the effectiveness of post operative analgesia like total post operative analgesic consumption and time to first request of analgesia are better if included.

• In the methodology session it says “VAS were categorized to define pain control as ‘adequate pain control’ (0-5) and ‘inadequate pain control’ (6-10).” do you have a reference for this statement

Result

Fair! But unless the points and factor suggested in the comment of the method session are incorporated, the quality of the result is questionable.

Discussion

Good!

Conclusion

The conclusion session didn’t provide a clear summary of the finding of the research. Furthermore, the conclusion session clearly seems a recommendation. It would be better if it is rewritten to show what exactly the finding of the research is.

6. PLOS authors have the option to publish the peer review history of their article (what does this mean?). If published, this will include your full peer review and any attached files.

Reviewer #1: **Yes: **Dr Carmen Power

Reviewer #2: No

---

## [Author Response · Author response to Decision Letter 0]

26 Mar 2022

RESPONSE TO REVIEWERS

Responses to Academic Reviewer

1. In the ethics statement please indicate the type of informed consent obtained from the parent or guardian of the minors included in the study (ie written, verbal).

Response: This has been done per the comment and the following added to the ethics statement

“For participants below 18 years of age, assent was documented but a verbal consent was obtained from the parent or guardian.”

2. In your Data Availability statement, you have not specified where the minimal data set underlying the results described in your manuscript can be found. PLOS defines a study's minimal data set as the underlying data used to reach the conclusions drawn in the manuscript and any additional data required to replicate the reported study findings in their entirety. All PLOS journals require that the minimal data set be made fully available…………………………..

Response: The minimal data set underlying the results described in the manuscript has been uploaded as Supporting Information File S1 Table (Assessment of Post-CS data). The sentence below has been added as the last sentence under ‘Data management and analysis’ under ‘Methods’

“The minimal dataset underlying the results described has been included as supporting information file S3 Table.”

Responses to Reviewer #1

1. Typos: ‘pan’ instead of ‘pain, p3

Response: The typographical error is acknowledged and has been rectified

2. P19 – movement was considered ‘presumably to breastfeed’ although women could move for other reasons post CS, such as visiting the bathroom, reaching for something such as a tissue or glass of water, or simply sitting up to eat a meal.

Response: The authors acknowledge the myriad reasons for which there could be movement although emphasis was somehow on breastfeeding to fulfil early breastfeeding initiation. The sentence in question has been modified to read as below;

“Movement, for any purpose including breastfeed, increased pain and erased associations found at rest within the same time period.”

3. The authors speak about the possible interference of participant pain with mother-baby bonding, yet they do not mention using a bonding scale. Therefore, this gap could be filled in follow up work or in some future research study.

Response: Exploring the deep dynamics of mother-baby bonding following CS was outside the scope of the study but we acknowledge the prospects of further investigating the subject. The following sentence has been added to the ‘Conclusion’

“The possible interference of post-CS pain with breastfeeding and mother-baby bonding, however short-lived, may also be explored in future studies by collecting data on periods of breastfeeding and using a bonding scale.”

4. Although they state that women breastfed their babies despite inadequate pain control, they do not specify exactly how many women managed to do this or if it was all the participants. Although, as traditionally 100% of women initiate breastfeeding in some African countries such as Tanzania, this may have been the case in Ghana too. This could be clarified for western readers in the discussion on p20.

Response: Barring any severe maternal clinical condition or neonatal asphyxiation that may hinder breastfeeding, all women undergoing CS in our setting breastfeed as early as possible just as the reviewer mentions is done in Tanzania. The sentence in question has been modified to reflect this and now reads as below;

“Despite the high prevalence of inadequate pain control at 6-12 hours with movement in the present study, it was observed during data collection that all the women still breastfed their infants.”

Responses to Reviewer #2

1. The title and the objectives of the study are not inline. For instance, the title included the burden of pain management but the main body of the document showed nothing regarding the burden of poor post cesarean delivery pain management.

Response: Burden, as used in the title, refers to the prevalence or proportion of women with adequate/inadequate perceived post-CS pain control at the defined periods of assessment. If we understand the reviewer correctly, the financial and social impact of the ‘burden’ was outside the scope of the study. To curtail any ambiguity however, the title has been revised to leave out the word ‘burden’ and now reads as;

“Assessment and determinants of acute post-caesarean section pain in a tertiary facility in Ghana”

The objectives part of the study is now aligned with the title and rephrased as;

“The study primarily sought to assess the adequacy of pain management post-CS, using a validated pain scale, from the perspective of women who had CS done in the second largest tertiary facility in Ghana as well as factors influencing this outcome. Secondarily, the prescription and serving habits of doctors and nurses respectively for post-CS analgesia were assessed.”

2. The introduction and method sessions of the abstract didn’t provide clear information about the problem and the methodology the researchers went through.

Response: The introduction and method sections of the abstract have been revised and nuanced to better reflect the problem driving the study and the methodology the researchers went through

The introduction now reads as;

“Caesarean sections (CS) feature prominently in obstetric care and have impacted positively on maternal / neonatal outcomes globally including Ghana. However, in spite of documented increasing CS rates in the country, there are no studies assessing the adequacy of post-CS pain control. This study assessed the adequacy of post-CS pain management as well as factors influencing this outcome. Additionally, post-CS analgesia prescription and serving habits of doctors and nurses were also described to help fill existing knowledge gaps. ” 

The methodology section of the abstract now reads as;

“Pain scores of 400 randomly selected and consenting post-CS women at a tertiary facility in Ghana were assessed at 6-12 hours post-CS at rest and with movement and at 24-36 hours post-CS with movement using a validated visual analog scale (VAS) from February 1, 2015 to April 8, 2015. Participant characteristics including age, marital status and duration of CS were obtained using pretested questionnaires and patient records review. Descriptive statistics were presented as frequencies and proportions. Associations between background characteristics and the outcome variables of adequacy of pain control at 6-12 hours post-CS at rest and with movement and at 24-36 hours post-CS with movement were analysed using Chi-square and Fisher’s exact tests and logistic regression methods. Adequate pain control was defined as VAS scores ≤5.”

3. Introduction

Well written, but in the objective part terms like adequacy are not measurable and vague

Response: To improve understanding of how adequacy is to be measured, the phrase ‘using a validated pain scale’ has been introduced. The part in question now reads as;

“The study primarily sought to assess the adequacy of pain management post-CS, using a validated pain scale, from the perspective of women who had CS done in the second largest tertiary facility in Ghana as well as factors influencing this outcome. In addition, the prescription and serving habits of doctors and nurses respectively for post-CS analgesia were assessed.”

4.The study design is not clearly described. It simply says prospective study. There are different prospective study types

Response: The study employed a cross-sectional design with repeated measures at fixed time intervals to assess changes that have taken place between surveys. The cross-sectional element stems from the fact that exposure and outcome were measured simultaneously while the measures at rest/movement at different time intervals of 6-12 hours and 24-36 hours post-CS underpin the element of repetition. Data analysis was aligned with this study design. The 1st sentence under the ‘Study area, study design and population’ has been modified to now read as;

“The study employed a cross-sectional design with repeated measures at fixed time intervals to assess changes in the perceived adequacy of post-CS pain control. It was conducted at the lying-in wards of the Obstetrics and Gynaecology Directorate of the Komfo Anokye Teaching Hospital (KATH) from February 1, 2015 to April 8, 2015.”

5. It would be good if the post-operative pain management protocol of the hospital was described.

Response: At the time of study conduct, there was on defined protocol for post-CS pain management in the hospital. Various surgeons used various combinations of opioids and non-steroidal anti-inflammatory drugs. There is thus no particular protocol that can be described. Such practices are rampant but can hardly be described as best practice and have been alluded to in the 2nd paragraph of the ‘Introduction’. Nevertheless, the following statement has been added in the 2nd paragraph under ‘Participant selection, study procedures and data collection tools’ under Methods to give a contextual background.

“No defined protocol or choice of medicines for post-CS analgesia existed at the time of study conduct and analgesia prescription was driven by availability and individual surgeon preference.”

6.The inclusion and exclusion criterias are not clearly stated. For instance, giving consent to participate in the study should not be considered as inclusion criteria. What about the American Society of Anesthesiologists (ASA) physical status of the patients for inclusion and exclusion criteria? Other possible inclusion and exclusion should be addressed.

Response: Giving informed consent, as part of inclusion criteria, has been taken off. Again, the investigators did not encounter the participants prior to surgery. They were recruited post-CS. Hence the ASA classification, which typically is pre-surgery, was not used to define inclusion/exclusion criteria. The reported inclusion/exclusion criteria are what were used for the study. The relevant section is now revised to read as below;

“Post-CS women of all ages in the lying-in wards at KATH were invited to participate in the study. Women were deemed eligible if they (i) had their CS done at KATH and (ii) were conscious and communicating by 6-12 hours after the surgery. No consideration was given to their physical status pre-surgery. For clients below 18 years, assent was obtained from the client and consent obtained from the attending guardian. Participants were excluded if there was a language barrier which could challenge communication. Information was extracted from the theatre register for the purposes of participant selection while socio-demographic, clinical characteristics and pain scores were obtained using patient folders and questionnaires after obtaining informed consent.”

7.The study didn’t consider other important factors which influence the severity of post cesarean delivery pain. For example, gestational age, type of local anesthetic used for spinal anesthesia, type of the post operative pain management modality used ( regional analgesia vs systemic analgesics), length of surgical incision, level of sensory block of the spinal anesthesia, quality of intraoperative analgesia and etc. were not addressed in the study. These factors are very critical for post cesarean delivery pain and analgesic quality assessment

Response: Only systemic analgesics for post-operative pain management are used in our set-up. Intraoperative analgesia is not routine. The other factors have been captured as potential subjects to be investigated in future studies in the “Conclusion”. It is captured as follows;

“….Also, the influence of other factors such as gestation, length of surgical incision and level of sensory block of the spinal anaesthesia can be further investigated in new studies that utilize parameters like total post-operative analgesic consumption in addition to the VAS for better assessment of the effectiveness of pain control.”

8.Other important parameters to assess the effectiveness of post operative analgesia like total post operative analgesic consumption and time to first request of analgesia are better if included.

Response: These are acknowledged but were not used in the study. Their use has been recommended for future studies in the “Conclusion”. Please see the response to (7) above.

9. In the methodology session it says “VAS were categorized to define pain control as ‘adequate pain control’ (0-5) and ‘inadequate pain control’ (6-10).” do you have a reference for this statement

Response: There is no reference for the statement. This categorization was arbitrary. The score of 5 in between was chosen as the border between ‘adequate’ and ‘inadequate’ pain contrl. This has been inserted in the 3rd paragraph of ‘Participant selection, study procedures and data collection tools’ under Methods and now reads as;

“……….and the pain scores by the VAS were categorized to define pain control as ‘adequate pain control’ (0-5) and ‘inadequate pain control’ (6-10). The middle score on the VAS was arbitrarily chosen as the border between adequate and inadequate pain control. 

10. Result

Fair! But unless the points and factor suggested in the comment of the method session are incorporated, the quality of the result is questionable.

Response: The comments raised regarding the methods section have been appropriately factored in.

11. Conclusion

The conclusion session didn’t provide a clear summary of the finding of the research. Furthermore, the conclusion session clearly seems a recommendation. It would be better if it is rewritten to show what exactly the finding of the research is

Response: Revisions have been made to reflect summaries of salient research findings

---

## [Decision Letter · Decision Letter 1]

12 May 2022

Assessment and determinants of acute post-caesarean section pain in a tertiary facility in Ghana.

PONE-D-21-20342R1

Dear Dr. Osarfo,

We’re pleased to inform you that your manuscript has been judged scientifically suitable for publication and will be formally accepted for publication once it meets all outstanding technical requirements.

Kind regards,

Carla Pegoraro

Division Editor

PLOS ONE

Reviewers' comments:

Reviewer's Responses to Questions

**Comments to the Author**

1. If the authors have adequately addressed your comments raised in a previous round of review and you feel that this manuscript is now acceptable for publication, you may indicate that here to bypass the “Comments to the Author” section, enter your conflict of interest statement in the “Confidential to Editor” section, and submit your "Accept" recommendation.

Reviewer #1: All comments have been addressed

2. Is the manuscript technically sound, and do the data support the conclusions?

Reviewer #1: (No Response)

3. Has the statistical analysis been performed appropriately and rigorously? 

Reviewer #1: (No Response)

4. Have the authors made all data underlying the findings in their manuscript fully available?

Reviewer #1: (No Response)

5. Is the manuscript presented in an intelligible fashion and written in standard English?

Reviewer #1: (No Response)

6. Review Comments to the Author

Reviewer #1: (No Response)

7. PLOS authors have the option to publish the peer review history of their article (what does this mean?). If published, this will include your full peer review and any attached files.

Reviewer #1: **Yes: **Carmen Power (PhD)

---

## [Editor Report · Acceptance letter]

16 May 2022

PONE-D-21-20342R1 

Assessment and determinants of acute post-caesarean section pain in a tertiary facility in Ghana. 

Dear Dr. Osarfo:

I'm pleased to inform you that your manuscript has been deemed suitable for publication in PLOS ONE. Congratulations! Your manuscript is now with our production department. 

Kind regards, 

on behalf of

Dr Carla Pegoraro 

Staff Editor

PLOS ONE